# PBRM1 Cooperates with YTHDF2 to Control HIF-1α Protein Translation

**DOI:** 10.3390/cells10061425

**Published:** 2021-06-08

**Authors:** Alena Shmakova, Mark Frost, Michael Batie, Niall S. Kenneth, Sonia Rocha

**Affiliations:** 1Centre for Gene Regulation and Expression, School of Life Sciences, University of Dundee, Dundee DD1 5EH, UK; alyona.shmakova@gmail.com; 2Institute of Systems, Molecular and Integrative Biology, University of Liverpool, Liverpool L69 7ZB, UK; Mark.Frost@liverpool.ac.uk (M.F.); M.Batie@liverpool.ac.uk (M.B.); Niall.Kenneth@liverpool.ac.uk (N.S.K.)

**Keywords:** PBRM1, HIF-1, SWI/SNF, YTHDF2, m6A, hypoxia

## Abstract

PBRM1, a component of the chromatin remodeller SWI/SNF, is often deleted or mutated in human cancers, most prominently in renal cancers. Core components of the SWI/SNF complex have been shown to be important for the cellular response to hypoxia. Here, we investigated how PBRM1 controls HIF-1α activity. We found that PBRM1 is required for HIF-1α transcriptional activity and protein levels. Mechanistically, PBRM1 is important for HIF-1α mRNA translation, as absence of PBRM1 results in reduced actively translating HIF-1α mRNA. Interestingly, we found that PBRM1, but not BRG1, interacts with the m6A reader protein YTHDF2. HIF-1α mRNA is m6A-modified, bound by PBRM1 and YTHDF2. PBRM1 is necessary for YTHDF2 binding to HIF-1α mRNA and reduction of YTHDF2 results in reduced HIF-1α protein expression in cells. Our results identify a SWI/SNF-independent function for PBRM1, interacting with HIF-1α mRNA and the epitranscriptome machinery. Furthermore, our results suggest that the epitranscriptome-associated proteins play a role in the control of hypoxia signalling pathways.

## 1. Introduction

The cellular response to decreased oxygen tension (hypoxia) is characterised by changes in the gene expression program primarily mediated by the hypoxia-inducible factor (HIF) family of transcription factors. HIF transcription factors are heterodimers composed of an oxygen-labile HIF-α subunit and a constitutively expressed HIF-1β subunit. In well-oxygenated cells, HIF-α subunits are targeted for proteasomal degradation by sequential proline hydroxylation and polyubiquitination mediated by proline hydroxylases (PHDs) and the Von Hippel–Lindau (VHL) E3 ubiquitin ligase, respectively [1]. In hypoxic cells, PHD enzymes are inhibited, leading to the stabilisation and accumulation of HIF-α isoforms. This results in active transcription of HIF target genes [2].

In order to facilitate the extensive changes in gene expression necessary for the cellular response to hypoxic stress, changes to the chromatin structure must take place for transcription factors such as HIF to access the promoter regions of its target genes. Chromatin exists in a spectrum of closed and open conformations determined by the density of nucleosomes, with active transcription requiring an open conformation of chromatin so that protein complexes can bind to the underlying DNA [3]. One mechanism of regulating chromatin structure is through the action of ATP-dependent remodelling complexes, which can result in either increased or decreased gene transcription [3]. There are four families of ATP-dependent remodelling complexes, imitation switch (ISWI), chromodomain helicase DNA-binding (CHD), INO80 and switch/sucrose non-fermentable (SWI/SNF). Each subfamily is specialised to preferentially achieve particular chromatin outcomes: assembly, access or editing, and has been shown to contribute to gene expression changes in response to hypoxic stress [3,4].

The levels and activity of SWI/SNF chromatin remodelling complexes are important for controlling gene expression changes in hypoxia [5]. SWI/SNF complexes are a family of large multisubunit complexes containing at least 8–12 subunits including an ATPase module, either BRG1 or BRM1, which exert a sliding force to displace nucleosomes to allow the binding of the transcriptional machinery to the promoters of target genes [6]. The function of SWI/SNF complexes is defined by the composition of subunits within each subcomplex with PBAF containing PBRM1, ARID2 and PFH10, while BAF complexes contain the ARID1A/B protein [6].

We have previously shown that the SWI/SNF components BAF155, BAF57 and BRG1 are required for HIF-1α mRNA expression, and their depletion impairs the HIF-dependent hypoxic response [7]. Indeed, not only is chromatin remodelling necessary for HIF-1α mRNA expression, but both BRG1 and BRM are required for full activation of HIF-1 and HIF-2 target gene expression in Hep3B, RCC4T and SW13 cells [8]. More recent work has shown overexpression of the SWI/SNF subunit, protein polybromo-1 (PBRM1), increases the levels of both HIF-1α and HIF-2α target genes (PHD2, GLUT3, BNIP3) [9], suggesting PBRM1 could regulate HIF-1α in a similar way to other SWI/SNF complex members. Interestingly, PBRM1 along with the HIF-1α and HIF-2α E3 ligase, VHL, is the most frequently mutated gene in clear cell renal cell carcinoma (ccRCC) [10].

PBRM1 is a large (193 kDa) multidomain protein containing six bromodomains which bind to acetylated histones, BAH domains which are protein-binding domains [11] and an HMG domain which has been shown to bind to the DNA minor groove [12]. Although PBRM1 is a core component of the PBAF complex, numerous studies have demonstrated nucleosome remodelling-independent functions of PBRM1 including localisation with kinetochores during mitosis [12], sister chromatid cohesion [13], DNA double-strand break repair [14] and interferon signalling [15,16]. Therefore, how PBRM1 levels and activity can alter the HIF-dependent hypoxic response is not yet clear.

Here, we investigated the involvement of PBRM1 in regulating HIF activity. We found that PBRM1 is required for full HIF activity and HIF-1α protein levels. However, this regulation occurs through promoting efficient translation of HIF-1α mRNA rather than through direct effects on HIF-1α gene transcription. This activity is supported by PBRM1’s ability to bind selectively to HIF-1α mRNA as well as the RNA-binding protein YTHDF2, which is also required for normal levels of HIF-1α protein in specific cellular backgrounds. This reveals a function for PBRM1 distinct from chromatin regulation and the rest of the SWI/SNF complex and suggests that the epitranscriptome and its machinery are important for the cellular response to hypoxia.

## 2. Materials and Methods

### 2.1. Cell Culture and Treatment

Cell lines were obtained from the European Collection of Cell Cultures. The cell lines were maintained in Dulbecco’s modified Eagle’s medium (DMEM) (Gibco/ThermoFisher, Paisley, UK) supplemented with 2 mM L-glutamine, 10% (*v*/*v*) foetal bovine serum (FBS) (Gibco/ThermoFisher, Paisley, UK), 50 units/mL penicillin (Lonza, Slough, UK) and 50 μg/mL streptomycin (Lonza, Slough, UK). HRE-luciferase reporter cell lines were maintained in these conditions supplemented with 0.5 µg/mL puromycin. All the cell lines were routinely tested for mycoplasma contamination using a MycoAlert kit from Lonza. Hypoxia treatments were performed in an InVivo 300 or InVivo2 hypoxia workstation (Baker Ruskinn, Bridgend, Wales) at 1% O_2_, 5% CO_2_ and 37 °C. The cells were treated with MG132 (Calbiochem/Millipore, Feltham, UK) at the final concentration of 20 μM, cycloheximide (Millipore, Feltham, UK) at the final concentration of 0.1 mg/mL, actinomycin D at the final concentration of 5 µg/mL and FG-4592 (Selleckchem/Stratech, Cambridge, UK) at the final concentration of 50 µM.

### 2.2. Transfection of siRNA and DNA

Transfections in HeLa, U2OS, H1299 and A549 were performed using INTERFERin (Polyplus, Illkirch, France) following the manufacturer’s instructions. The HEK293 cells were transfected with 0.12 M CaCl_2_ and HEPES buffered saline (0.156 M NaCl, 0.375 M Na_2_HPO_4_, 10 mM HEPES were mixed with water to the final volume of 400 µL and added to the cells). The cell culture medium was changed 24 h following transfection and the cells were harvested 48 h following transfection. GFP-PBRM1 was a kind gift of Prof. Jessica Downs and was described in [14]. For DNA transfections, 1 ug of plasmid DNA was transfected per well of a six-well plate. Small interfering RNA (siRNA) oligonucleotides (Eurofins) were transfected at 27 nM. The siRNA oligonucleotide sequences were as follows: control, CAGUCGCGUUUGCGACUGG; PBRM1_A GAAGAAAGCAUUAAGGUAU; PBRM1_B UCAGGACGUCTCAUTAGCGAA; YTHDF2 AAGGACGTTCCCAATAGCCAA.

### 2.3. Immunoblotting

The cells were lysed in the RIPA buffer (50 mM Tris-HCl (pH 8), 150 mM NaCl, 1% (*v/v*) NP40, 0.25% (*w/v*) Na deoxycholate, 0.1% (*w/v*) SDS, 10 mM NaF, 2 mM Na_3_VO_4_ and 1 tablet/10 mL cOmplete, Mini, EDTA-free protease inhibitors (Roche)), centrifuged for 10 mins at 13,000 rpm at 4 °C, collecting the supernatant for analysis. SDS-PAGE and immunoblotting were performed using standard protocols. The following primary antibodies were used for immunoblotting: HIF-1α (610958, BD Biosciences, Workingham, UK), HIF-2α (sc-13596, Santa Cruz, Dallas, TX, USA), HIF-1β (3718, Cell Signalling, Leiden, Holland), β-Actin (3700, Cell Signaling, Leiden, Holland/60009-1-Ig, Proteintech, Manchester, UK), PBRM1 (ABE70, Millipore, Feltham, UK), YTHDF2 (24744-1-AP, Proteintech, Manchester, UK/80014, Cell Signaling, Leiden, Holland), GFP (2956, Cell Signaling), BRG1 (Santa Cruz, Dallas, TX, USA, sc-17796). Following incubation with a horseradish peroxidase-conjugated secondary antibody (Cell Signalling Technology, Leiden, Holland), chemiluminescence (Pierce/ThermoFisher, Paisley, UK) was detected. All the figures are representative of a minimum of three independent experiments.

### 2.4. Luciferase Assay

The cells lines stably expressing the HRE-luciferase reporter were transfected and/or treated as indicated before lysis in a passive lysis buffer. Luciferase assays were performed according to the manufacturer’s instructions (Promega, Southampton, UK) and activity was measured using a Lumat LB 9507 luminometer (EG&G Berthold, Bad Wildbad, Germany). The results were normalised according to protein concentration and reported as a percentage of control.

### 2.5. Quantitative PCR (qPCR)

RNA from the HeLa cells was extracted using a peqGOLD total RNA kit (Peqlab, Bishop’s Waltham, UK), from the HEK293 cells—using a Direct-Zol RNA MiniPrep kit (Zymo Research, Irvine, USA) according to the manufacturers’ protocols. RNA was reverse-transcribed using a Quantitect Reverse Transcription kit (Qiagen, Manchester, UK). Real-time PCR was performed using the Brilliant II SYBR Green kit (Stratagene/Agilent, Stockport, UK) on an Mx3005P qPCR machine (Stratagene/Agilent, Stockport, UK). Levels of mRNA were calculated based on averaged Ct values and normalized to β-actin mRNA levels. The following primers were used for qPCR: *HIF-1α* forward CATAAAGTCTGCAACATGGAAGGT, *HIF-1α* reverse ATTTGATGGGTGAGGAATGGGTT, *HIF-1β* forward CAAGCCCCTTGAGAAGTCAG, *HIF-1β* reverse GAGGGGCTAGGCCACTATTC, *β-actin* forward CCCAGAGCAAGAGG, *β-actin* reverse GTCCAGACGCAGGATG, *HK2* forward AGCCCTTTCTCCATCTCCTT, *HK2* reverse AACCATGACCAAGTGCAGAA, *PHD2* forward GAAAGCCATGGTTGCTTGTT, *PHD2* reverse TGTCCTTCTGGAAAAATTCG, *PHD3* forward CTTGGCATCCCAATTCTTGT, *PHD3* reverse ATCGACAGGCTGGTCCTCTA, GLUT1 forward TCAAAGGACTTGCCCAGTTT, *GLUT1* reverse GATTGGCTCCTTCTCTGTGG, *VEGF* forward AGCTGCGCTGATAGACATCC, *VEGF* reverse CTACCTCCACCATGCCAAGT, *BNIP3* forward GCCCACCTCGCTCGCAGACAC, *BNIP3* reverse CAATCCGATGGCCAGCAAATGAGA.

### 2.6. Polysome Profiling

The HeLa cells were transfected in 10-cm plates. Translation was stimulated by changing the media 1.5 h before lysis. Cycloheximide in the amount of 0.1 mg/mL was added 10 mins before lysis. The cells were washed twice with 5 mL ice-cold PBS containing 0.1 mg/mL cycloheximide. The cells were collected in 300 µL PBS with cycloheximide, and the PBS was removed by centrifugation. The cells were resuspended in 550 µL polysome extraction buffer (15 mM Tris-HCl (pH 7.5), 15 mM MgCl_2_, 300 mM NaCl, 1% Triton X-100, 1 mM DTT, 0.1 mg/mL cycloheximide and RNAsein (Sigma)). Cellular debris was pelleted by centrifugation at 17,000× *g* for 1 min at 4 °C. The supernatant in the amount of 500 µL was added to 10 mL 10–50% sucrose gradients (with the extraction buffer without Triton X-100). The gradients were centrifuged using an SW41 rotor at 223,000× *g* for 2 h at 4 °C. The gradient was collected in 1 mL fractions. RNA was extracted from the solution with an RNAeasy extraction kit (Qiagen) according to the manufacturer’s protocol. RNA was converted to cDNA using a Quantitect Reverse Transcription kit (Qiagen) on a SureCycler 8800 (Agilent) and an Mx3005P qPCR machine. Brilliant II SYBR Green (Agilent) was used to perform qPCR on an Mx3005P (Agilent).

### 2.7. Coimmunoprecipitation

The cells were lysed in the RIPA buffer and the lysate was diluted with an equal volume of the buffer (50 mM Tris-HCl (pH 8.0), 0.5 mM DTT, 20% glycerol, 5 mM NaF, 500 mM Na_3_VO_4_ and 1 EDTA-free protease inhibitor tablet per 10 mL buffer). The cell lysate in the amount of 500 μg per condition was incubated with 2 μg PBRM1 antibody (Bethyl Labs. Montegomety, AL, USA, A301-591A) or 2 μg IgG control (Sigma, Gillinham, UK) and rotated at 4 °C overnight. Immune complexes were captured with 20 μL protein G Sepharose beads (Generon. Slough, UK) by incubating with rotation for 1.5 h at 4 °C. The beads were washed three times with PBS, then the beads were boiled with 20 µL SDS loading buffer.

### 2.8. RNA Immunoprecipitation

Either the HeLa or H1299 cells were grown in 15-cm plates, washed twice with ice-cold PBS, harvested in ice-cold PBS and pelleted. Of these cells, 10% were used for each condition. The cells were lysed with 100 µL RIP lysis buffer (Magna RIP, Millipore, Feltham, UK) and snap frozen. The cells were thawed and centrifuged at 15,000× *g*. The supernatant was diluted 10-fold with the Magna-RIP wash buffer. The magnetic beads were coated with 5 µg of the appropriate antibody for 30 mins at room temperature. The beads were incubated with the cell extract with rotation overnight at 4 °C. The beads were then washed six times with the RIP wash buffer. After setting aside 10% of the beads for IP control, RNA was eluted by treatment with proteinase K according to the manufacturer’s instructions, then purified by phenol–chloroform extraction followed by an 80% ethanol wash. RNA was finally dissolved in water, converted to cDNA and quantified with qPCR comparing to 500 ng of the input.

### 2.9. In Vitro RNA Binding Assay

For RNA bait preparation, DNA duplexes containing T7 promoter sequence (5′-TAATACGACTCACTATAG-3′) followed by either the HIF-1α UTR (CAGTGCTGCCTCGTCTGA) or the luciferase mRNA control sequence (GGAGCCCCTGCTAACGACATTTACAACGAG) were generated by PCR. RNA was generated with transcription by the T7 RNA polymerase according to the manufacturer’s instructions (Thermo Fisher, Paisley, UK). Each 20-µL reaction was spiked with 200 ng DNA template and 0.5 µM biotin-labelled CTP. The reaction mixture was then treated with 2 units of DNase and incubated at 37 °C. The RNA was purified using an RNA purification column (PeqGOLD, Peqlab, Bishop’s Waltham, UK). The secondary structure was ensured by heating to 90 °C followed by 2 min of incubation on ice and adding 50 µL RNA structure buffer (10 mM Tris-HCl (pH 7.4), 100 mM KCl, 10 mM MgCl_2_, 1 mM DTT, 0.4 U/µL RNase inhibitor, Sigma, Gillinham, UK). RNA was conjugated to beads by mixing 100 µL of each RNA with 20 µL streptavidin agarose beads (Sigma, Gillinham, UK, S1638) and incubating with mixing for 30 min at 4 °C. The beads were washed twice with 5 mM Tris-HCl (pH 7.4), 1 M NaCl. Ten-centimetre plates of PC-3 cells per pulldown were lysed in a buffer (10 mM HEPES, pH 7.9, 1.5 mM MgCl_2_, 10 mM KCl, 0.1 mM PMSF and 0.5 mM DTT, supplemented with an EDTA-free protease inhibitor tablet (Roche, Welwyn Garden city, UK)). The cells were homogenised on ice using 10 strokes of a Dounce homogeniser, then centrifuged at 16,100× *g* for 15 min at 4 °C. The supernatant in the amount of 1 mg was incubated with each of the bead-conjugated RNAs for 3 h at 4 °C, then washed three times with a wash buffer (25 mM Tris-HCl (pH 7.4), 150 mM KCl, 2 mM MgCl_2_, 1 mM DTT, 0.5% IGEPAL, 1 mM PMSF, 0.4 U/µL RNAse inhibitor, supplemented with an EDTA-free protease inhibitor tablet (Roche, Welwyn Garden city, UK)). The proteins present were analysed by SDS-PAGE and immunoblotting.

### 2.10. Recombinant Protein Expression and In Vitro Pulldown Assay

The *E. coli* BL21 DE3 (Invitrogen) cells were transformed with plasmids containing 6xHis-tagged YTHDF2 (1–384, 384–579 or 1–579) and PBRM1 (BAH1 and 2 967–1287 or HMG 1394–1462 domains). The secondary cultures in the amount of 250 mL were inoculated and grown to OD_600nm_ 0.6, then induced with 0.5 mM IPTG and incubated with shaking overnight at 18 °C. The cells were pelleted, resuspended in a lysis buffer (0.1 M Tris, pH 8, 0.5 M NaCl, 5% glycerol, 20 mM imidazole, 1 mM β-ME, protease inhibitor tablet), then sonicated for 60 s in 10-s pulses at 80% amplitude on ice. The lysed cells were centrifuged at 40,000× *g* and the clarified lysate was mixed with 40 µL Ni-NTA affinity resin for 4 h at 4 °C. The beads were washed in the lysis buffer for 15 mins, then 1 mg of the HeLa cell lysate was added and the mixture was incubated at 4 °C overnight with rotation. The beads were then washed three times with a wash buffer (50 mM Tris pH 7.4, 200 mM NaCl, 5% glycerol, 0.1% Tween-20, 1 mM DTT). The beads were boiled with an SDS loading buffer and analysed by Western blotting.

## 3. Results

### 3.1. PBRM1 Is Required for HIF Transcriptional Activity

As previously discussed, SWI/SNF chromatin remodeller activity has been associated with HIF-dependent and -independent responses in hypoxia [7,8,9]. Furthermore, several SWI/SNF complex members are often mutated in cancer. Of note, PBRM1 is often mutated in renal cancer, where the HIF system plays an important role in the oncogenic process [10]. To specifically examine the effects of PBRM1 in the control of HIF activity, siRNA oligonucleotides were used to deplete PBRM1 in a variety of cell lines containing an HRE-luciferase reporter prior to exposure to hypoxia. PBRM1 levels were effectively depleted by two independent siRNAs as determined by immunoblotting (Figure 1A). PBRM1 depletion resulted in a significant decrease in HIF activity in all the cell lines tested (Figure 1A). We then investigated if this was also reflected at the level of endogenous HIF-dependent targets using qPCR. We were able to detect significant decreases in the levels of HK2, PHD2 and PHD3 mRNA in the HeLa cells (Figure 1B) but not of GLUT1 mRNA levels (Figure 1B). In the HEK293 cells, PBRM1 depletion resulted in decreased mRNA levels of all the HIF targets we investigated (Figure 1C). These results suggest that knockdown of PBRM1 inhibits hypoxia-induced HIF activity in multiple cell lines.

### 3.2. PBRM1 Is Required for HIF-1α Protein Expression in Normoxia and Hypoxia

HIF activity is primarily regulated by the availability of the oxygen-sensitive HIF-α subunits. This is dependent on the degradation of HIF-α by the proteosome. To investigate if PBRM has a role in regulating their abundance, levels of HIF-1α and HIF-2α were measured by immunoblotting in hypoxic cells in which PBRM1 levels were depleted using siRNA. Interestingly, depletion of PBRM1 specifically reduced the levels of hypoxia-induced HIF-1α (Figure 2A, Appendix A) but did not reduce the levels of the HIF-2α (Figure 2A, Appendix A) or HIF-1β subunits (Figure 2A). Conversely, PBRM1 overexpression increased the levels of hypoxia-induced HIF-1α but, again, had little effect on other HIF subunits (Figure 2B). We next investigated if inhibiting the proteosome would alter PBRM1-dependent reduction of HIF-1α protein levels. Analysis of PBRM1 depletion in the HeLa and HEK293 cells showed that HIF-1α levels could not be restored by treating cells with the proteosome inhibitor MG132 (Figure 2C). This was also the case when we co-depleted proteosomal component RPN11 with PBRM1 (Appendix A). Consistent with this observation, we were not able to detect changes in the stability of HIF-1α protein when PBRM1 was depleted (Appendix A). We were able to show that overexpression of PBRM1 led to increased levels of HIF-1α in the presence or absence of MG132 (Figure 2D). These results were similar to the observed effects following depletion of the core SWI/SNF subunits we had previously investigated in U2OS cells [7], suggesting a role in transcriptional regulation via BRG1. To confirm this hypothesis, we analysed the role of PBRM1 in the cells that have impaired BRG1, A549 and H1299 lung cancer cells [17]. Surprisingly, PBRM1 depletion in these cells resulted in similar reduction in HIF-1α levels following MG132 treatment (Figure 2E). These results indicated that PBRM1 regulation of HIF-1α occurs through a mechanism independent of BRG1.

### 3.3. PBRM1 Binds Selectively to HIF-1α mRNA and Promotes Polysome Processing

Our analyses of how PBRM1 controls HIF-1α suggest a mechanism that differs from that of SWI/SNF components BAF155, BAF57 and BRG1, regulation of HIF-1α transcription [7]. To investigate this, we examined HIF mRNA levels in the Hela cells after PBRM1 siRNA depletion. HIF-1α, HIF-2α and HIF-1β transcript levels all increased when PBRM1 was depleted (Figure 3A). Knockdown of BRG1 led to a reduction in HIF-1α mRNA level as expected (Figure 3B). Furthermore, PBRM1 depletion did not change HIF-1α mRNA in the HEK293 (Figure 3C) and H1299 cells (Figure 3D). We ruled out PBRM1 depletion decreasing HIF-1α mRNA stability (Appendix A). Clearly, PBRM1 does not regulate HIF activity through transcriptional regulation but has a function different from that of other SWI/SNF complex members.

It is known that PBRM1 can bind to DNA as part of its function on the PBAF complex [12], but we investigated whether it could bind to RNA as a mechanism for regulating the expression of HIF-1α. Using an RNA immunoprecipitation (RIP) assay, we found that PBRM1 can bind to HIF-1α mRNA (Figure 3E) as well as to mRNAs of other hypoxia-responsive HIF target genes (Appendix A), revealing a novel function of PBRM1. This RNA binding capacity was confirmed with an in vitro RNA binding assay demonstrating that PBRM1 can bind to HIF-1α mRNA via the 5′ UTR (Figure 3F). This RNA binding demonstrates specificity as PBRM1 was unable to bind to a control mRNA sequence.

As PBRM1 depletion could decrease HIF-1α protein levels but not transcript levels, we next investigated HIF-1α translation. Global protein synthesis, as well as translation rates of individual mRNAs, can be measured by polysome profiling, a technique in which free ribosomes can be separated from mRNA-bound ribosomes (polysomes) on a sucrose gradient. Quantitative PCR analysis of polysome-associated HIF-1α mRNA revealed that PBRM1 depletion caused a decrease in HIF-1α mRNA association with actively translating ribosomes consistent with a decrease in HIF-1α translation efficiency (Figure 3G,H). Importantly, no change was detected in the levels of polysome-associated mRNAs for HIF-2α and HIF-1β (Appendix A). These data suggest that HIF-1α mRNA translation is sensitive to levels of PBRM1.

### 3.4. PBRM1 but Not BRG1 Can Bind to the m6A-Binding Protein YTHDF2

Due to our unexpected observations that PBRM1 controlled HIF-1α levels through translational control, we interrogated the PBRM1 interactome using IP-MS (data not shown) to identify known translational regulators and RNA binding proteins as interactors of PBRM1. Among these, we found YTHDF2, which can recognise and bind to m6A-methylated RNA [18]. We were able to validate this observation using Western blotting, identifying that PBRM1 binds YTHDF2 in normoxia and reduced binding in hypoxia (Figure 4A). Interestingly, treatment with a PHD inhibitor (FG-4592) only slightly reduced the interaction between PBRM1 and YTHDF2 at four hours of treatment (Figure 4B). Using bacteria-expressed recombinant YTHDF2 as bait and the total HeLa cell lysate, we were able to detect a robust interaction with PBRM1 (Figure 4C). However, when the YTHDF2 protein was expressed in two parts, this interaction was either abolished (N-terminal part) or significantly reduced (C-terminal part). We were also able to determine that PBRM1 can interact with YTHDF2 via its HMG domain, while expression of BAH1 and 2 domains was insufficient to bind YTHDF2 (Figure 4D). Furthermore, treatment with RNAse A had no effect on the binding between YTHDF2 and the HMG domain of PBRM1 (Figure 4E), indicating that the interaction between YTHDF2 and PBRM1 is not dependent on the presence of RNA, but is more likely mediated by protein–protein interaction. Consistent with our data presented above, immunoprecipitation of the BRG1 complex, which also contains PBRM1, does not bind YTHDF2, suggesting that PBRM1 has functions outside of the canonical PBAF complex (Figure 4F).

### 3.5. PBRM1 Is Required for YTHDF2 Binding to HIF-1α mRNA and for HIF-1α Protein Expression

Our data suggest that HIF-1α RNA could be m6A methylated. To investigate this further, publicly available datasets were interrogated [19,20,21,22,23]. Our analysis revealed that HIF-1α has indeed been identified both in human and mouse tissues as being m6A-modified, and also as bound by YTHDF2 in mouse cells [20] (Appendix A). We next determined if we could validate this observation in our cell system. Indeed, we confirm that HIF-1α mRNA is m6A-modified and can be bound by YTHDF2 (Figure 5A). To determine the interplay between YTHDF2 and PBRM1 in regulating HIF-1α, we depleted PBRM1 by siRNA and assessed HIF-1α mRNA transcript bound to YTHDF2 (Figure 5B) Our results indicated that YTHDF2 binding to HIF-1α mRNA requires the presence of PBRM1 (Figure 5B). To investigate the functional significance of PBRM1 binding to YTHDF2, we determined if YTHDF2 was required for HIF-1α protein expression. Depletion of YTHDF2 decreased HIF-1α protein following MG132 treatment in both the HeLa and H1299 cell lines (Figure 5C,D), demonstrating their requirement for HIF-1α protein expression.

Taken together, our data demonstrate a previously unknown function of PBRM1, outside the SWI/SNF complex and in cooperation with the epitranscriptome, in the control of HIF-1α mRNA fate.

## 4. Discussion

Here, we investigated the role of PBRM1, a component of the SWI/SNF complex, in the regulation of the HIF response. We used a variety of cancer and non-cancer cell lines in this study. Components of the SWI/SNF chromatin remodeller are frequently mutated in cancer [24], and as such, understanding their role in conditions of hypoxia, a stress prevalent in malignancies, is of importance. We found that PBRM1 is required for HIF transcriptional activity through promoting HIF-1α protein synthesis. Unlike the core members of the SWI/SNF complex [7], PBRM1 can promote efficient translation of HIF-1α mRNA though polysome processing. This activity is supported by PBRM1’s ability to bind selectively to HIF-1α mRNA along with the m6A RNA-binding protein YTHDF2, which is also required for normal levels of HIF-1α protein. Our data reveal a function for PBRM1 that is separate from its involvement in the SWI/SNF complex as PBRM1 is able to control HIF-1α protein in the absence of a functional ATPase. Furthermore, we found that HIF-1α is modified by m6A, suggesting that RNA methylation is required for full efficiency of the hypoxia-inducible pathway.

There is a precedent for RNA binding activity within the SWI/SNF complex with the catalytic subunits BRG1 and BRM interacting with lncRNA and mRNA [25,26]. Some RNA processing has been shown to occur co-transcriptionally [27] with the PPB1 subunit of RNA polymerase II acting as a platform for recruiting machinery for 5’-end capping, splicing, polyadenylation and histone methylation [28,29]. Chromatin structure can influence the RNA fate as H3K4me3 has been shown to recruit the RNA-capping machinery [30]. SWI/SNF has been shown to affect the elongation rate of RNA Pol II, influencing alternative exon inclusion [31]. PBRM1 contains a HMG domain, which is known to bind DNA, and in some cases preferentially bind RNA [32,33]. We found that PBRM1 is able to bind to HIF-1α mRNA in cells and in vitro. This demonstrates a new function of PBRM1 which is independent of its role in a catalytic SWI/SNF complex as PBRM1 depletion also affects HIF-1α levels in H1299 cells which are deficient in BRG1 [17,34]. It would therefore be of interest to conduct an unbiased analysis using RIP-seq, for example, to determine how widespread this new PBRM1 function is. An unusual function of PBRM1 was very recently reported [35]. PBRM1 was found to bind to methylated microtubules. This binding was necessary to recruit the SWI/SNF members to the mitotic spindle and ensure genomic stability [35] Our results similarly show that PBRM1 binds methylated sites, though in RNA, however, this seems to be independent of the other SWI/SNF components.

In hypoxia, global translation is reduced, regulated at the stage of initiation [36], posing a paradox with the increases in protein levels of hypoxia-inducible genes. Several mechanisms are proposed to contribute to the selective translation of HIF-1α mRNA in hypoxia (reviewed in [37]). Our analysis revealed that HIF-1α mRNA is m6A-modified. This has also been seen in an unbiased genome-wide analysis of this modification [21,23]. More than 150 different posttranscriptional modifications have been identified on cellular RNAs [38], with the most abundant being the m6A methylation within the coding region or UTRs of the RNA, particularly in mRNAs [18]. RNA methylation is “written” by the methyltransferase complex METTL3/14, WTAP, RBM15/15B and KIAA1429, “erased” by the demethylases FTO and ALKBH5 and recognised or “read” by YTHDF1/2/3 [18]. RNA m6A modification regulates RNA splicing [39], processing [39], decay [40] and translation [20,41]. N6-methyladenosine acts as a binding site for several reader proteins, including YTHDF2. YTHDF2 binding is normally associated with increased RNA degradation [42]. However, under stress conditions such as heat shock, YTHDF2 translocates to the nucleus and protects 5′ UTR m6A, which promotes cap-independent translation of target genes [21]. A similar mechanism may operate for HIF-1α mRNA, as we found that YTHDF2 becomes more nuclear-located in response to hypoxia (Appendix A). This suggests a role for YTHDF2 in locating to recently transcribed RNA in the nucleus and contributing to hypoxia-selective translation in cells.

Our data suggest that PBRM1 is required for efficient HIF-1α protein translation. However, in hypoxia, we observed a reduced interaction between PBRM1 and YTHDF2, suggesting that, potentially, another m6A reader might come into play. Further analysis investigating m6A levels in hypoxia and PBRM1 interaction could provide answers to these remaining questions. However, the importance of PBRM1 is still present in hypoxia, as reduced HIF-1α protein translation in the absence of PBRM1 is still observed under this condition. This indicates that reductions in transcription [7,37] and translation in normoxia impact HIF-1α protein levels in hypoxia.

Although the analysis of the epitranscriptome has not been conducted in hypoxia, there are existing links between these two aspects. The m6A eraser proteins, FTO and ALKBH5, are 2-oxoglutarate dioxygenases, meaning they use 2-oxoglutarate, iron and oxygen as cofactors. Although their oxygen affinity might be too high to participate in the hypoxia response [43], ALKBH5 has been shown to be a target of HIF-1α [44]. This suggests that, indeed, hypoxia can influence the epitranscriptome in a cell. Furthermore, studies of liver cancer have shown that hypoxia can increase the levels of m6A in mRNA [45]. More recently, METTL3, one of the m6A writers, has been shown to promote hypoxia-induced hypertension in pulmonary arteries [46]. Our data demonstrating that HIF-1α is m6A-modified adds to the hypothesis that, indeed, the epitranscriptome is an integral part of the cellular response to hypoxia.

## Figures and Tables

**Figure 1 cells-10-01425-f001:**
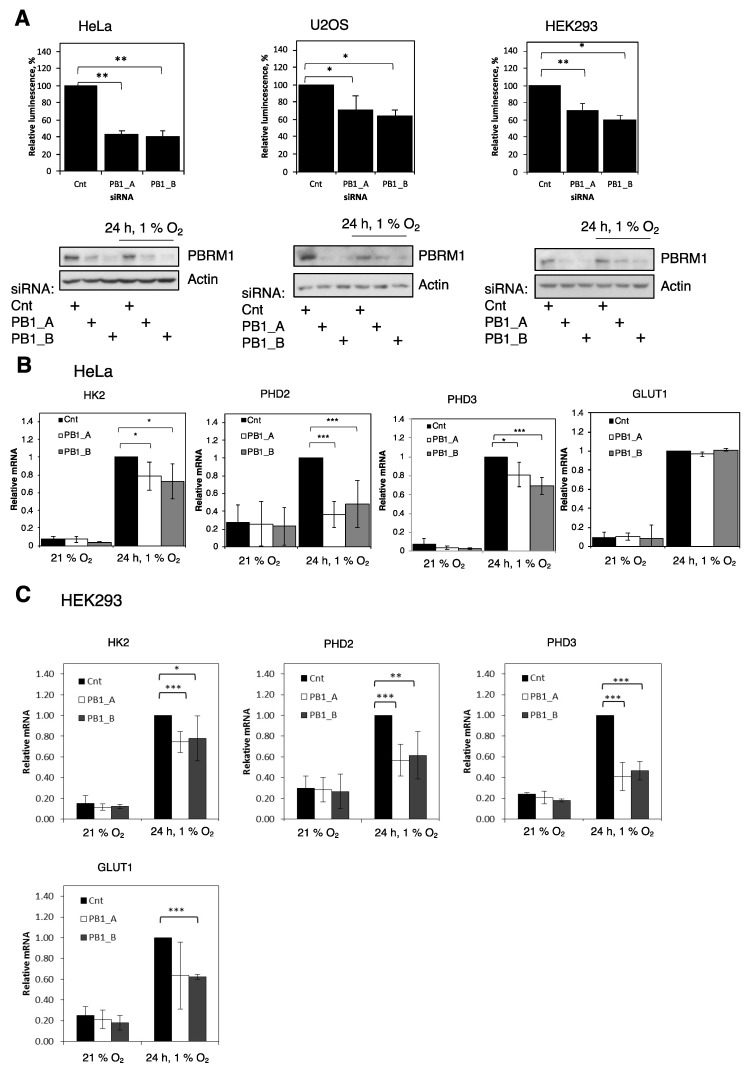
PBRM1 is required for HIF transcriptional activity. (**A**) The HeLa, U2OS and HEK293 HRE-luciferase reporter cell lines were transfected with either control (Cnt) or PBRM1 (PB1) siRNAs for 48 h prior to lysis and exposed to 1% O_2_ for 24 h prior to lysis. Cell lysates were assayed for luciferase activity and Western-blotted. The graphs display the means and standard deviations (SD) of a minimum of three independent biological replicates with ANOVA significance indicated (* *p* ≤ 0.05, ** *p* ≤ 0.01). +, signifies specific treatment. (**B**) The HeLa cells and (**C**) the HEK293 cells were transfected and treated with hypoxia as before. Total RNA was extracted, and mRNA levels were assessed by qPCR, normalised to actin and compared to control siRNA. The graphs depict the means and SD of a minimum of three independent biological replicates with ANOVA significance indicated (* *p* ≤ 0.05, ** *p* ≤ 0.01. *** *p* ≤ 0.001).

**Figure 2 cells-10-01425-f002:**
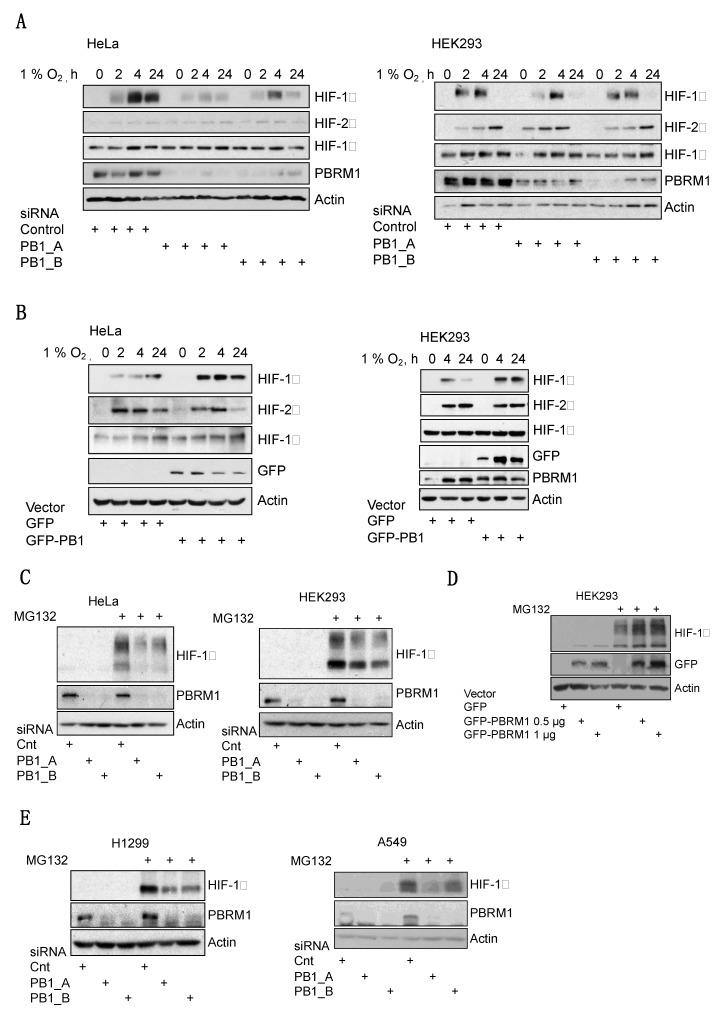
PBRM1 is required for HIF-1α protein levels independently of proteasomal degradation. (**A**) The HeLa and HEK293 cells were transfected for a total of 48 h with either control or PBRM1 (PB1) siRNAs before exposure to treatments with 1% O_2_ of different duration, +, signifies specific treatment. Whole-cell lysates were analysed by Western blotting to assess HIF isoform protein levels. (**B**) The HeLa and HEK293 cells were transfected with 1 µg of either the GFP empty vector or a plasmid containing GFP-PBRM1. The cells were then exposed to 1% O_2_ for treatments of different duration. Whole-cell lysates were analysed by Western blotting to assess HIF isoform protein levels. (**C**) The HeLa and HEK293 cell lines were transfected with control or PBRM1 siRNAs, then treated with DMSO or MG132 for 3 h before lysis. Whole-cell lysates were analysed by Western blotting to assess HIF-1α protein levels. (**D**) The HEK293 cells were transfected with 1 µg of either the GFP empty vector or increasing amounts of a plasmid containing GFP-PBRM1. The cells were then treated with DMSO or MG132 for 3 h before lysis. Whole-cell lysates were analysed by Western blotting to assess HIF-1α protein levels. (**E**) The H1299 and A549 cells were transfected with control or PBRM1 siRNAs, then treated with DMSO or MG132 for 3 h before lysis. Whole-cell lysates were analysed by Western blotting to assess HIF-1α protein levels.

**Figure 3 cells-10-01425-f003:**
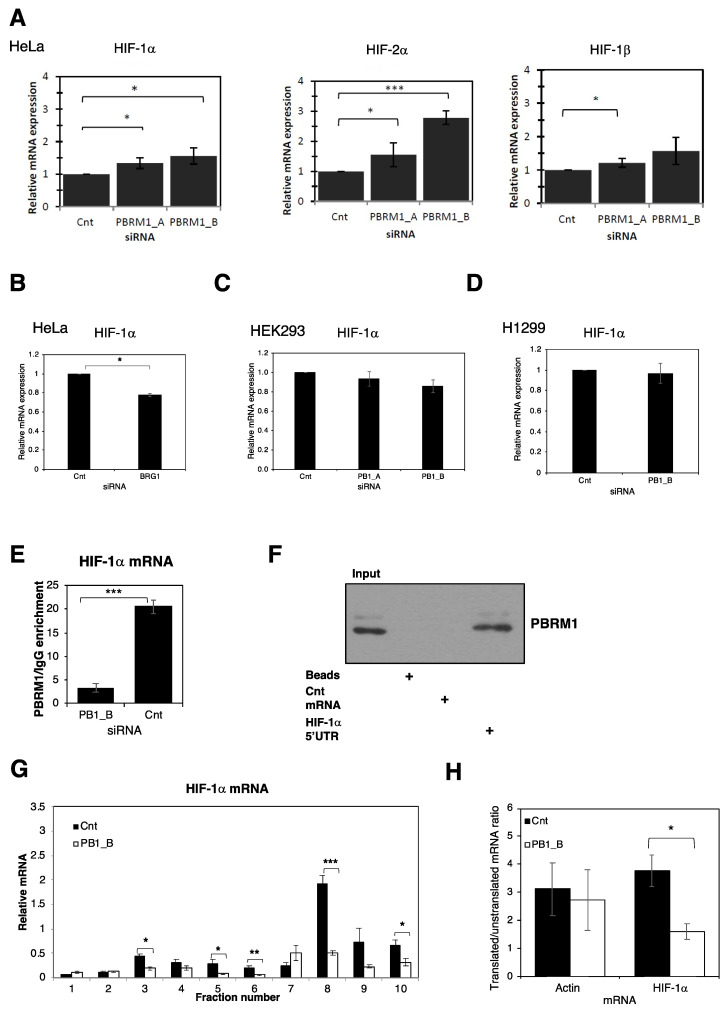
PBRM1 binds HIF-1α mRNA and regulates HIF-1α translation through polysome processing. (**A**) The HeLa cells were transfected with control and BRG1 siRNAs for 48 h; then, the cells were lysed and the mRNA levels of HIF transcripts were assessed by qPCR as compared to actin levels. The means and standard deviations are plotted with ANOVA significance indicated (* *p* ≤ 0.05, ** *p* ≤ 0.01, *** *p* ≤ 0.001). (**B**) The HeLa, (**C**) HEK293 and (**D**) H1299 cells were transfected with control and PBRM1 siRNAs for 48 h; then, the cells were lysed and the mRNA levels of HIF transcripts were assessed by qPCR as compared to actin levels. The means and standard deviations are plotted with ANOVA significance indicated (* *p* ≤ 0.05, ** *p* ≤ 0.01, *** *p* ≤ 0.001). (**E**) The HeLa cells were transfected with either control or PBRM1 siRNAs; then, 48 h later, they were lysed and incubated with the PBRM1 antibody coupled to magnetic beads. After washing, any RNA present was converted to cDNA and analysed by qPCR using primers for HIF-1α. The means and standard deviations are plotted with ANOVA significance indicated (* *p* ≤ 0.05, ** *p* ≤ 0.01, *** *p* ≤ 0.001). (**F**) Biotinylated control and HIF-1α RNA bait were conjugated to beads, then incubated with 1 mg HeLa cell lysate. The beads were washed and PBRM1 protein levels were assessed by Western blotting. +, signifies specific treatment. (**G**) The HeLa cells were transfected with control or PBRM1 siRNA and lysed 48 hours later. The clarified HeLa cell lysate was added to a 10–50% sucrose gradient and centrifuged at 223,000× *g*, then collected in fractions. RNA was extracted and levels of HIF-1α were determined by qPCR for each fraction and reported relative to actin levels. The means and standard deviations are plotted with ANOVA significance indicated (* *p* ≤ 0.05, ** *p* ≤ 0.01, *** *p* ≤ 0.001). (**H**) The ratios of mRNA between polysomal (7–10) and monosomal (1–6) fractions are shown for actin and HIF-1α transcripts with and without mRNA knockdown of PBRM1. The means and standard deviations are plotted with ANOVA significance indicated (* *p* ≤ 0.05, ** *p* ≤ 0.01, *** *p* ≤ 0.001).

**Figure 4 cells-10-01425-f004:**
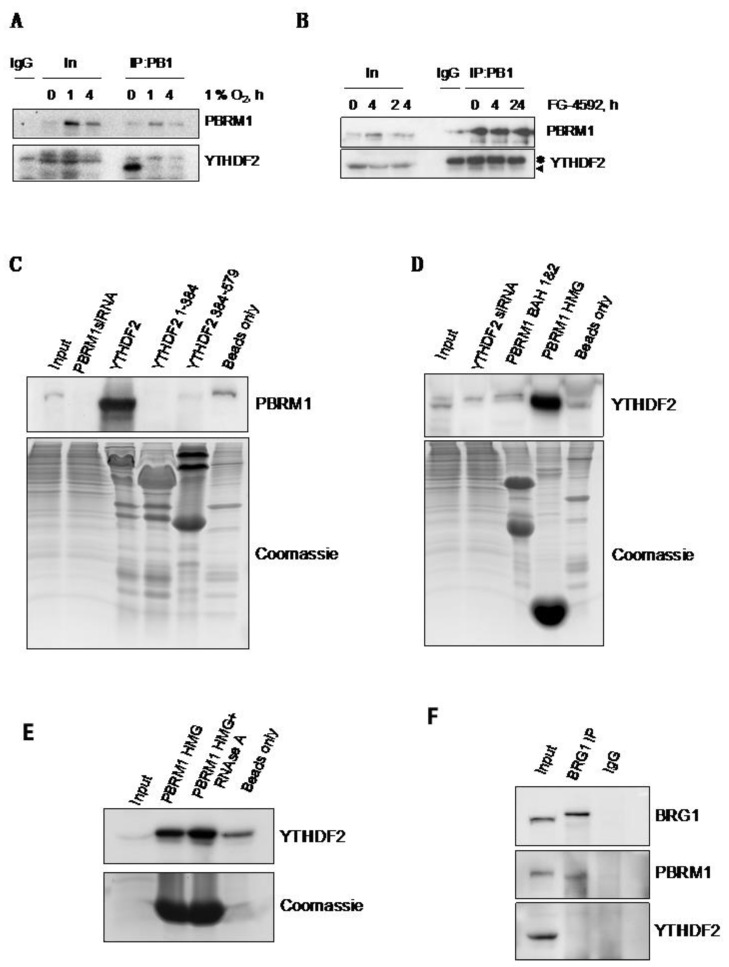
PBRM1 but not BRG1 can bind to the m6A-binding protein YTHDF2 in cells and in vitro. (**A**) The HeLa cells were exposed to 1% O_2_ treatments of different duration. The cell lysate in the amount of 500 µg was immunoprecipitated with either the PBRM1 antibody or the IgG control. Levels of PBRM1 and YTHDF2 were determined by Western blotting. Input represents 10% of the protein. (**B**) The HeLa cells were treated with 50 µM of FG-4592 for the indicated periods of time. The cell lysate in the amount of 500 µg was immunoprecipitated with either the PBRM1 antibody or the IgG control. Levels of PBRM1 and YTHDF2 were determined by Western blotting. Input represents 10% of the protein. Note: * indicates a heavy chain, the arrow indicates the YTHDF2 signal. (**C**) Recombinant His-tagged domains of YTHDF2 or (**D**) PBRM1 bound to Ni-NTA beads were incubated with 1 mg of the total HeLa cell lysate overnight at 4 °C and then washed; protein levels were analysed by Western blotting. Input represents 10% of the protein. (**E**) Recombinant His-tagged PBRM1 HMG domains (1394–1462) bound to Ni-NTA beads were incubated with 1 mg of the total HeLa cell lysate overnight at 4 °C with or without 50 µg of RNAse A and then washed; YTHDF2 levels were analysed by Western blotting. Input represents 10% of the protein. (**F**) The HeLa cell lysate in the amount of 500 µg was immunoprecipitated with either the BRG1 antibody or the IgG control. Levels of BRG1 and YTHDF2 were determined by Western blotting. Input represents 10% of the protein.

**Figure 5 cells-10-01425-f005:**
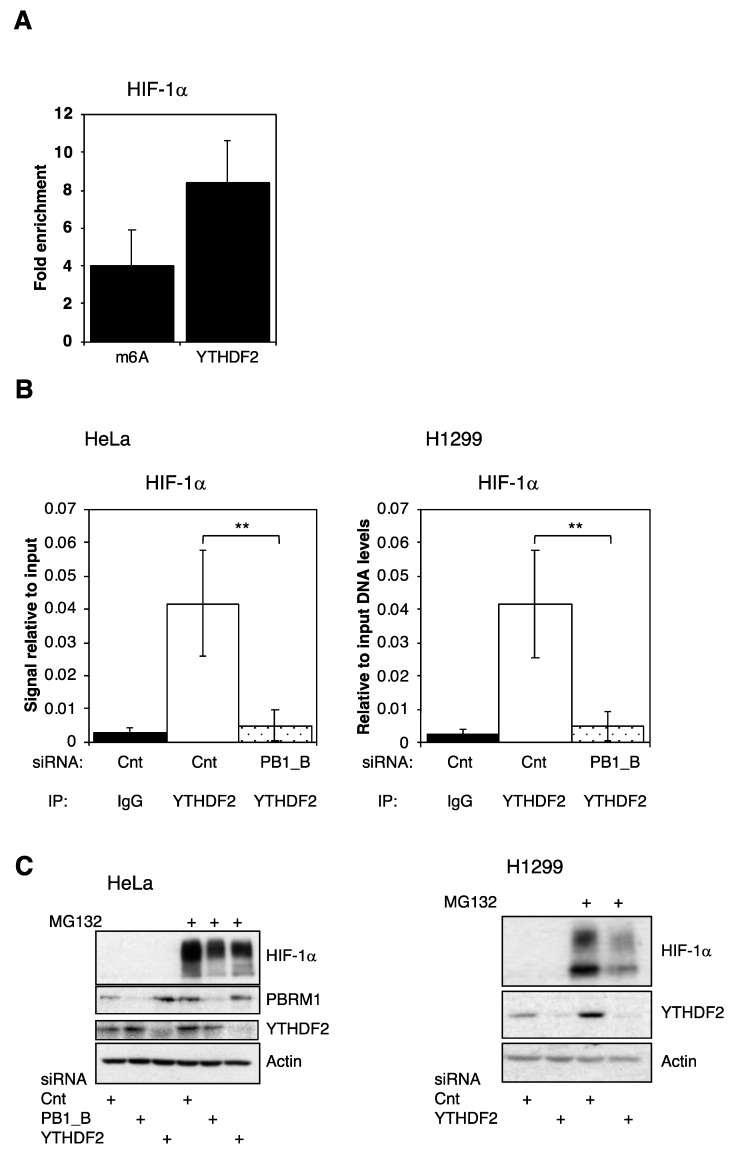
HIF-1α mRNA is m6A-modified and bound by YTHDF2. (**A**) The HeLa cells were lysed and incubated with either m6A or YTHDF2 antibodies; then, bound HIF-1α mRNA was analysed by qPCR. (**B**) The HeLa and H1299 cells were then transfected with either control or PBRM1 siRNAs and incubated with the IgG control or YTHDF2 antibodies before analysis of HIF-1α mRNA levels. The means and standard deviations are plotted with Student’s *t*-test significance indicated (** *p* ≤ 0.01). (**C**) The HeLa or H1299 cells were transfected with control, PBRM1 or YTHDF2 siRNAs and either DMSO or MG132 for 3 h; then, the cells were lysed and protein levels were determined by Western blotting. +, signifies specific treatment.

## Data Availability

The additional data such as biological replicates is available upon request.

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
