# Peer review of "PBRM1 Cooperates with YTHDF2 to Control HIF-1α Protein Translation"

_cells, 2021, doi:10.3390/cells10061425_

Round 1
Reviewer 1 Report
In this study Shmakova A et al. claim that PBRM1 is required for optimal HIF1a expression because it acts as an inducer of HIF1a mRNA translation. Authors propose that PBRM1 do this action by favoring the binding of YTHDF2 to m6A methylated-modified HIF1a RNA. Data are very interesting. However the following comments should be addressed.
Major comments
- - In Figure 4A authors show that hypoxia markedly reduces the interaction between PBRM1 and YTHDF2. Do authors suggest that hypoxic cells induces HIF1a protein expression despite of the fact that HIF1a RNA translation is reduced as a consequence of the disruption of PBRM1-YTHDF2 complex? Authors should discuss about this point.
- - If PBRM1 does not interact with YTHDF2 in hypoxic cells (Figure 4A), why HIF1a levels are still reduced in PRBM1-silenced cells or YTHDF2-silenced cells? Authors should discuss about this point.
- - Is PBRM1-YTHDF2 interaction reduced by exposure to PHDs inhibitors or in PHDs-silenced cells?
4.- In Figure 2A it might be suggested to include another cell line (or cell lines) with higher levels of endogenous HIF2a expression. This will help to confirm that HIF2a is not affected upon PRBM1 silencing.
Minor points
- - Do HIF1a protein expression increases in PBRM1-deficient Vhl-deficient renal cell carcinoma cells lines upon PBRM1 restoration?
Author Response
Major comments
- - In Figure 4A authors show that hypoxia markedly reduces the interaction between PBRM1 and YTHDF2. Do authors suggest that hypoxic cells induces HIF1a protein expression despite of the fact that HIF1a RNA translation is reduced as a consequence of the disruption of PBRM1-YTHDF2 complex? Authors should discuss about this point.
We have included additional discussion regarding this point: “Our data suggests that PBRM1 is required for efficient HIF-1a protein translation. However, in hypoxia we observed a reduced interaction between PBRM1 and YTHDF2, suggesting that potentially another m6A-reader might come into play. Further analysis investigating m6A levels in hypoxia and PBRM1 interaction could provide the answer to these remaining questions.”
- - If PBRM1 does not interact with YTHDF2 in hypoxic cells (Figure 4A), why HIF1a levels are still reduced in PRBM1-silenced cells or YTHDF2-silenced cells? Authors should discuss about this point.
We have included this point in the discussion. PBRM1 is necessary for translation of HIF-1a in normal conditions, so cells in hypoxia start with a lower translational level for HIF-1a: “However, the importance of PBRM1 is still present in hypoxia, as reduced HIF-1a protein translation in the absence of PBRM1 is still observed under this conditions. This indicates that reductions in transcription [42,43] and translation in normoxia impact on HIF-1a protein levels in hypoxia.”
- - Is PBRM1-YTHDF2 interaction reduced by exposure to PHDs inhibitors or in PHDs-silenced cells?
We have now included this new data in Figure 4B, treatment with PHD inhibitor only slightly reduces PBRM1 interaction with YTHDF2 and only after 24h of treatment.
4.- In Figure 2A it might be suggested to include another cell line (or cell lines) with higher levels of endogenous HIF2a expression. This will help to confirm that HIF2a is not affected upon PRBM1 silencing.
As suggested we have included HEK293 cells and in supplementary 786-O cells (VHL null and HIF-1a null renal cancer cell), showing no reduction in HIF-2alpha levels.
Minor points
- - Do HIF1a protein expression increases in PBRM1-deficient Vhl-deficient renal cell carcinoma cells lines upon PBRM1 restoration?
We have only been able to obtain this data once, this is because when cells are transfected with PBRM1, the cells die. So technically we could not really perform this experiment. We managed to analyse one experiment but we are not confident with this result due to the health of the transfected cells.

Reviewer 2 Report
The authors have submitted their article entitled PBRM1 cooperates with YTHDF2 to control HIF-1α protein translation. There are however some important concerns and comments that should be addressed by the authors in order to consider the manuscript for publication.
- In all figures, fonts and labels for statistical analysis are too small, particularly in the bar graphs. In addition, Hif1-α becomes Hif1-□ in figure 1, 2, and 3…please check carefully.
- A graphical abstract should be provided by the authors. It should summarize the contents of the article in a concise, pictorial form designed to capture the attention of a wide readership.
- line243…p should be italic and a p is deleted.
- Western blotting images provided in the study only show one experiment. In general, western blotting should be at least triplicated and analyzed statistically.
- HeLa, U2OS, H1299, and A549 were conducted in the study. Why these cell lines used should be discussed.
- In figure 1A, PBRM1 levels should be analyzed in both normal and hypoxia conditions.
Author Response
We thank the reviewer for their constructive comments and suggestions

Round 2
Reviewer 1 Report
Authors have addressed satisfactorily my comments.